# Emergent Peptides of the Antifibrotic Arsenal: Taking Aim at Myofibroblast Promoting Pathways

**DOI:** 10.3390/biom13081179

**Published:** 2023-07-28

**Authors:** Zhen Liu, Xinyan Zhang, Yanrong Wang, Yifan Tai, Xiaolin Yao, Adam C. Midgley

**Affiliations:** 1State Key Laboratory of Medicinal Chemical Biology, Key Laboratory of Bioactive Materials for the Ministry of Education, College of Life Sciences, Nankai University, Tianjin 300071, China; 2120211248@mail.nankai.edu.cn (Z.L.); 2120221530@mail.nankai.edu.cn (X.Z.); 2120221441@mail.nankai.edu.cn (Y.W.); 1120190539@mail.nankai.edu.cn (Y.T.); 2School of Food and Biological Engineering, Shaanxi University of Science and Technology, Xi’an 710021, China; yaoxiaolin@sust.edu.cn

**Keywords:** fibrosis, myofibroblast, antifibrotic, peptides, peptide therapy

## Abstract

Myofibroblasts are the principal effector cells driving fibrosis, and their accumulation in tissues is a fundamental feature of fibrosis. Essential pathways have been identified as being central to promoting myofibroblast differentiation, revealing multiple targets for intervention. Compared with large proteins and antibodies, peptide-based therapies have transpired to serve as biocompatible and cost-effective solutions to exert biomimicry, agonistic, and antagonistic activities with a high degree of targeting specificity and selectivity. In this review, we summarize emergent antifibrotic peptides and their utilization for the targeted prevention of myofibroblasts. We then highlight recent studies on peptide inhibitors of upstream pathogenic processes that drive the formation of profibrotic cell phenotypes. We also briefly discuss peptides from non-mammalian origins that show promise as antifibrotic therapeutics. Finally, we discuss the future perspectives of peptide design and development in targeting myofibroblasts to mitigate fibrosis.

## 1. Introduction

Prevalent tissue fibrosis is associated with end-stage diseases in multiple organs. Myofibroblasts are the cells with the foremost culpability for the elevated production and deposition of the disorganized collagen-rich extracellular matrix (ECM) that forms the pathological hallmark of fibrosis [1]. The chronic activation and prolonged presence of myofibroblasts within tissues are related to the progression of fibrosis [2,3]. The current consensus is that there are prerequisites for the generation of mature α-smooth muscle actin (α-SMA)-positive myofibroblasts, including stimulation of the transforming growth factor (TGF)-β1 pathway, generation of extracellular stress forces from ECM-mechanoreceptor interactions, and the presence of phenotypic priming biomolecules (extra domain A–fibronectin, EDA-FN; hyaluronan, HA; etc.). Myofibroblasts are heterogeneous in their origin, but the cell differentiation pathways and signaling cascades that drive differentiation are common among the precursor cells. Thus, there has been a surge in research activity directed towards developing therapeutics that target these shared myofibroblast-promoting pathways. We refer interested readers to our previous review that goes into extensive detail on the function and formation of myofibroblasts [4].

Amidst the numerous classes of biologics that have been considered for use as antifibrotic tools, peptides have emerged as promising candidates due to their inherent natural origin, biocompatibility, and bioactivity, as well as improved diversity in modification potential and production efficiency compared to their larger protein counterparts. Peptides designed with biomimicry in mind aim to convey activities akin to growth factors, receptor agonists, competitive binders, and protein antagonists. Alternatively, the identification of cleaved protein generation of in situ peptide neoepitopes [5,6] and intermediate peptide forms with elevated potency [7] as modes of eliciting beneficial actions has inspired investigations into the use of peptide neoepitopes and peptide intermediate forms as endogenous therapeutics. In this review, we collate recently identified peptide solutions that interact with and modulate key myofibroblast-promoting pathways, namely the TGF-β/Smad signaling pathways and the ECM/mechanotransduction pathways (Figure 1). Additionally, we summarize promising anti-fibrotic peptides that enact regulation over pathways that are not exclusive to myofibroblasts but are upstream and involved in promoting tissue fibrosis. We then briefly highlight peptides from non-mammalian origins that show promising potential for mitigating fibrotic processes. We finalize the review with a detailed discussion regarding the future perspectives of antifibrotic peptides, including sources of inspiration the concomitant development of technologies to aid in peptide discovery, design, and development as antifibrotic therapeutics.

## 2. Direct Targeting of Myofibroblast Pathways with Peptides

### 2.1. Peptides Targeting the TGF-β Pathway

The TGF-β signaling pathway is considered the consensus canonical pathway in myofibroblast differentiation [8,9,10,11]. As such, the TGF-β1 and -β2 pathways have long been popular targets for antifibrotic interventions. Several components and contributors to successful TGF-β pathway activation, signal transduction, and downstream effects have presented opportunities for peptide intervention and interruption of signaling at different cellular locations (Table 1).

In the extracellular space and amongst the ECM, the glycosaminoglycan (GAG) and heparan sulfate (HS) can bind TGF-β1 and -β2 to potentiate their biological activities and receptor interactions [18]. Poosti et al. [12] demonstrated that truncated proteins that contained the highly positively charged COOH-terminal region of C-X-C chemokine ligand 9 (CXCL9) could bind GAGs with high affinity. They developed a synthetic peptide mimetic of the CXCL9 COOH-terminal, CXCL9(74-103), which was capable of competing with free TGF-β1 to bind HS. The ability to interrupt TGF-β1/HS complexation was dependent on the presence of two GAG-binding domains in CXCL9(74-103). In unilateral ureteral obstruction (UUO) mouse models of renal fibrosis, CCXL9(74-103) was delivered continuously over the course of 7 days by subcutaneously implanted osmotic pumps and resulted in the inhibition of pro-fibrotic markers in the UUO kidneys, including α-SMA, vimentin, fibronectin, collagen type III, TGF-β1, and matrix metalloproteinase (MMP)-9. In addition, CXCL9(74-103) lessened inflammation, as determined by reduced expression of F4/80^+^ macrophages and the monocyte-attractant CCL2.

Upon TGF-β binding, TGF-β receptor type 2 (TGFβRII) dimerizes with TGF-β receptor type 1 (TGFβRI) on the fibroblast cell membrane, which enables TGFβRII phosphorylation of the TGFβRI kinase domain. Subsequently, phosphorylation of Smad2 and Smad3 and their oligomerization with co-Smad4 result in transcription or co-transcription factor activity in the nucleus and regulation of pro-fibrotic gene expression [4]. In the following sections, we summarize emergent peptides that target the TGF-β/Smad2/3 signaling pathway. Klotho is a single-pass transmembrane co-receptor of fibroblast growth factor 23 (FGF23) that is downregulated in aging or injured kidneys. Yuan et al. [13] developed a synthetic Klotho-derived peptide 1 (KP1), which contains the highly homologous and conserved 30 amino acid KL1 domain of Klotho. In the presence of KP1, TGF-β1 stimulated rat renal fibroblasts failed to differentiate into myofibroblasts. The mechanism of KP1 action was determined to be the competitive blockade of TGFβRII, thereby preventing TGF-β1 binding, dimerization of TGFβRII/TGFβRI, and downstream activation and nuclear translocation of Smad2/3. Tail vein injection of KP1 into unilateral renal ischemia-reperfusion injury (IRI) and UUO mouse models of renal fibrosis reciprocallyd inhibited TGFβRII activation and expression in vivo. Renal function was protected, and fibrosis was reduced without the initiation of dysregulated blood phosphorus and calcium metabolism that has been associated with elevated serum expression of Klotho. In addition, KP1 restored endogenous expression of Klotho, reduced the presence of UUO-induced renal F4/80^+^ macrophages, and blocked the infiltration of F4/80^+^ CD3^+^ macrophages.

C-type natriuretic peptide (CNP) induces the generation of the secondary messenger 3′,5′-cyclic guanosine monophosphate (cGMP), activation of protein kinase G-1a (PKG-1a), which in turn phosphorylates activated pSmad3 to prevent the translocation of the pSmad3/Smad4 heterodimer to the nucleus [19,20] and promotes proteasomal degradation of Smad3 [21]. Chen et al. [7] indicated that C53, a 53-amino acid intermediate form of pro-CNP, activated the particulate guanylyl cyclase B (pGC-B) receptor but elicited a more robust cGMP production response than CNP. The increased resistance of C53 against rapid degradation and catabolism by neprilysin provided sustained activity in vitro and resulted in anti-fibrotic actions in human cardiac and renal fibroblasts. M10 is a 10-amino acid peptide derived from the C-terminal cytoplasmic tail of the mesenchymal-epithelial transition factor (MET) receptor. Li et al. [14] demonstrated that M10 had anti-fibrotic effects during the early and late stages of silica-induced pulmonary fibrosis in mouse models. The authors then partially explored the underlying mechanisms in vitro. M10 was detected within both the cell cytoplasm and nuclei, and further investigation revealed its inhibition of Smad2 phosphorylation. M10 was suggested to have the capability to reverse silica-induced epithelial-to-mesenchymal transition (EMT) in epithelial cells and decrease TGF-β1 dependent activation of fibroblasts.

Hepatocyte growth factor splice variant NK1 (HGF/NK1) and bone morphogenetic protein 7 (BMP7) are two growth factors with demonstrated potent anti-fibrotic activities [22,23]. Huang and colleagues [15] employed H-RN peptide, derived from the kringle 1 domain present in HGF/NK1, to show in vitro inhibition of TGF-β2/Smad2/3-dependent induction of epithelial-mesenchymal transition (EMT) in lens epithelial cells, preventing their acquisition of myofibroblastic markers. H-RN also prevented TGF-β2/Smad-independent activation of Akt, mTOR, and p70S6K. The precise molecular mechanism of anti-fibrotic action and extent of HGF receptor MET’s role remain to be elucidated. The BMP7/Smad1/5/8 pathway competes with Smad2/3 for co-Smad4 binding, thereby interfering with pro-fibrotic gene transcription [24]. Salido-Medina et al. [16] evaluated the anti-fibrotic effects of BMP7-derived peptides, THR123 and THR184, in mouse models of transverse aortic constriction/release-induced cardiac remodeling and fibrosis. THR123 and THR184 bound BMP receptor 1A and increased phosphorylation of Smad1/5/(8)9 in the left ventricle (LV), which in turn rescued ventricular hypertrophy and dysfunction. Moreover, delayed administration of THR184 indicated its ability to prevent progressive remodeling and partially reverse LV dysfunction from pressure overload. Darmawan et al. [17] opted to use the adiponectin-derived peptide, ADP355, and demonstrated its antifibrotic effect on keloid fibroblasts and xenotransplanted keloid tissues. ADP355 inhibited procollagen expression and attenuated the activation of Smad3 while inducing the activation of AMP-activated protein kinase (AMPK), which has previously been described to interfere with Smad3 recruitment to gene promoters [25].

### 2.2. Peptides Targeting the ECM and Mechanotransduction

Mechanotransduction is required for myofibroblast maturation and terminal differentiation. Fibroblasts are mechanosensitive cells and perceive external stress forces through fibronexi and focal adhesion structures formed at the interface between the ECM, cell membrane receptors, and the cell cytoskeleton [26,27,28,29]. ECM substrate stiffness or rigidity promotes the conversion of extracellular ECM forces to intracellular signaling and contractile tension generation by the cell cytoskeleton. Elevated ECM microenvironment stiffness results in the incorporation of α-SMA into cytoplasmic β-actin stress fibers, granting myofibroblasts enhanced contractability [4]. Interruption of external-to-intracellular mechanotransduction signaling by preventing ECM assembly, ECM-receptor interactions, and downstream signaling associated with cytoskeletal reassembly can serve as suitable targets for peptide-based therapies (Table 2).

A collagen-rich ECM has increased stiffness, promotes myofibroblast differentiation, and contributes to the further excess production of collagens that contribute to the detriment of functional tissue. Thus, encouraging collagen breakdown rather than synthesis could be an effective means to reduce ECM substrate stiffness. B7-33 is a relaxin family peptide receptor 1 (RXFP1) agonist and derivative of the human gene-2 (H2)-relaxin hormone, which has been widely documented for its antifibrotic effects [34]. In a 2021 study by Bhuiyan et al. [30] on the utilization of an automated platform for collagen quantification by second-harmonic generation, the authors continuously administered B7-33 peptide to UUO mouse models via subcutaneously implanted micro-osmotic pumps over the course of 7 days. Interestingly, B7-33 treatment increased collagen fiber counts but ameliorated UUO-induced interstitial collagen fiber thickness, which prevented renal fibrosis. The changes were explained by B7-33’s induction of MMP-2 and suppression of tissue inhibitor of metalloproteinase (TIMP)-1 protein expression, suggesting a facilitation of collagen fiber breakdown. More recently, Alam et al. [35] demonstrated that B7-33 reciprocated the cardioprotective and antifibrotic effects of recombinant H2 relaxin hormone (RLX, Serelaxin) in mouse models of isoprenaline-induced cardiomyopathy. Treatment with B7-33 or RLX reduced the presence of myofibroblasts and the expression of TGF-β1 in the myocardium.

The endostatin neoepitope is formed in situ by the cleavage of collagen XVIII. A peptide derivative of the endostatin C-terminus, known as E4, was the subject of earlier studies showing its remarkable antifibrotic effects on dermal and pulmonary models of fibrosis [5]. In a recent study by Sharma et al. [6], the E4 mechanism of action was determined to be via binding to the urokinase-type plasminogen activator receptor (uPAR), which activated the urokinase and plasminogen pathways. Subsequently, increased MMP-1 and -3 activity resulted in an enhancement of the degradation of the collagen I-enriched ECM. Additionally, E4 promoted the production of urokinase-type plasminogen activator (uPA) and HGF while inhibiting the production of plasminogen activator inhibitor-1 (PAI-1). Furthermore, in ex vivo lung tissue cultures, E4 induced MMP-1 activity and reduced collagen I, fibronectin, and PAI-1. Indeed, previous studies also indicated the roles of uPAR in attenuating myofibroblast functions, and these included proliferation, recruitment, and the subsequent progression of renal fibrosis by regulating extracellular signal-regulated kinase (ERK) signaling and reducing the extracellular accumulation of PAI-1 [36], which are also well-documented downstream targets of TGF-β1 signaling [11,37,38].

The extra domain A-containing fibronectin splice variant (EDA-FN) is detectable during tissue repair and fibrosis [4]. EDA-FN promotes myofibroblast differentiation by orchestrating latency-associated peptide (LAP) mitigation of TGF-β1 activation, increasing ECM stress–strain tension, and activating mechanotransduction via integrin signaling pathways. In our own studies, we identified the integrin α4β1 (VLA-4) receptor site for EDA-FN and designed a synthetic polypeptide (AF38Pep) that competitively bound EDA-FN to prevent its association with VLA-4 and activation of downstream effectors [31]. Incorporation of AF38Pep into polymeric wound dressings facilitated its controlled release into rabbit ear models of hypertrophic scarring [32]. AF38Pep successfully prevented myofibroblast differentiation, attenuated excessive deposition of disorganized collagen types I and III, and inhibited hypertrophic scar formation, which resulted in improved quality of dermal wound repair. Interestingly, the peptide exerted anti-proliferation activity against activated fibroblasts, myofibroblasts, and keloid fibroblasts but did not affect the proliferation rate of inactivated fibroblasts. This finding indicated a potential barrier functionality against hyperproliferative cells, offering an alternative route towards achieving regulated growth of invasive mesenchymal cell phenotypes. Xu et al. [33] employed the short LSKL peptide to occupy the LAP-binding site of thrombospondin-1 (TSP-1) to antagonize TSP-1-mediated activation of latent TGF-β in mechanically-induced rat tail models of hypertrophic scarring. LSKL attenuated PI3K/AKT/mTOR signaling and significantly attenuated hypertrophic scar features, such as thickened dermis and distorted collagen alignment.

## 3. Peptide Mediators of Auxiliary Fibrosis-Promoting Mechanisms

In addition to targeting myofibroblast-promoting mechanisms, peptide-based interventions have been targeted at pathways commonly associated with driving fibrotic diseases to preemptively restrict the initiation of fibrogenic processes. In this section, we provide a brief overview of recently developed and employed peptides that target auxiliary processes, often upstream of fibrosis pathogenesis but not necessarily directly associated with the cellular mechanisms of myofibroblasts.

Inflammation represents an important process governing the development and progression of fibrosis. Intermedin (IMD) is a paracrine- and autocrine-secreted cardiovascular regulatory peptide with demonstrated functions in protecting against myocardial fibrosis. Zhang et al. [39] investigated the proteolytic cleavage product of prepro-IMD, designated IMD_1–53_, and demonstrated its non-selective binding and activation of the calcitonin receptor-like receptor (CRLR)/receptor activity modifying protein (RAMP) complexes to induce cAMP/PKA activation, which in turn inhibited inositol-requiring enzyme 1α (IRE1α) and NOD-like receptor family pyrin domain containing 3 (NLRP3) inflammasome activation, thereby attenuating inflammatory cytokines and ameliorating Angiotensin II (AngII)-induced myocardial fibrosis in rat models. Cyclic helix B peptide (CHBP) was produced by thioether cyclization and stabilization of the QEQLERALNSS amino acid sequence derived from erythropoietin [40]. CHBP demonstrated strong binding capacity to the tissue protective receptor (TPR) and exerted its anti-fibrotic activity in mouse models of UUO-induced renal fibrosis by inhibiting activation of the NLRP3 inflammasome. Serelaxin (RLX) has received extensive attention from the cardiovascular research community because of its promise for treating acute heart failure. Sassoli et al. [41] summarized the antifibrotic actions and highlighted the potential scope of RLX for treating fibrotic diseases in their detailed review. In a recent study, RLX was shown to inhibit the TGF-β1/IL-1β axis via a nNOS-TLR-4-NLRP3 inflammasome-dependent mechanism in cardiac myofibroblasts [42], which may be a mechanism that contributes to explaining the anti-inflammatory mechanisms observed in cardiomyopathy models treated with RLX and its derivative peptide, B7-33 [35]. Thymosin β4 (Tβ4) is a 43 amino acid polypeptide mediator of multiple cellular functions, including the sequestration of monomeric actin (G-actin). Xiang et al. [43] fabricated Tβ4-releasing fibrous scaffolds, which were capable of regulating pro-inflammatory macrophages and controlling the expression of TGF-β1 and fibroblast activation. In subcutaneous injury defect mouse models, the implantation of Tβ4-releasing fibrous scaffolds inhibited the formation of fibrosis during tissue repair.

Multiple receptors besides TGFβRII/I have been shown to contribute to pro-fibrotic cell activation and activity. The CSP (DGIWKASFTTFTVTKYWFYR) and CSP7 (FTTFTVT) peptide derivatives of caveolin-1 were shown to elicit anti-fibrotic effects through inducing microRNA-34a expression of p53, which subsequently inhibited the expression of sirtuin 1 (SIRT1), increased p53 expression, and led to blockade of platelet-derived growth factor receptor (PDGFR)-β activation and prevention of pulmonary fibrosis in bleomycin-injury mouse models [44].

Angiotensin converting enzyme (ACE), neutral endopeptidase (NEP), and aminopeptidase N (APN) are responsible for the generation of vasoactive peptides that regulate vasoconstriction, vasodilation, and natriuresis. Hence, these three enzymes may serve as attractive targets for regulating hypertension and preventing downstream fibrogenic processes. Savitha et al. [45] synthesized N-methylated peptide inhibitors (F-N(Me)H-L, V-N(Me)F-R, and R-N(Me)V-Y) against ACE, NEP, and APN using their respective physiological substrates. Evaluation of the anti-hypertensive effect of these peptides using a rat model of dexamethasone-induced hypertension indicated that administration of F-N(Me)H-L and a cocktail of all three peptides significantly reduced systolic blood pressure compared to a dexamethasone-only control group. In addition to anti-hypertensive effects, treatments with the N-methylated peptides had anti-hypertrophic and anti-fibrotic effects that were reflected by decreased levels of collagen deposition in heart and kidney tissues.

Insulin-like growth factor-1 (IGF-1) and the activation of its receptor have been implicated in phosphoinositide 3-kinase (PI3K)/Akt signaling, cardiomyocyte survival, and cardiac hypertrophy. Moreover, IGF-1 was suggested to be a critical regulator of AngII-associated cardiac fibrosis [46]. A study by Shafiq et al. [47] developed polycaprolactone/collagen type I cardiac patches that simultaneously eluted substance P (SP) and insulin-like growth factor-1C (IGF-1C) peptide (GYGSSSRRAPQT). The developed patches were transplanted into left anterior descending artery ligation-induced myocardial infarction mouse models over a course of 14 days. Mice that received peptide-eluting cardiac patches maintained heart function and attenuated adverse cardiac remodeling. Although antifibrotic mechanisms were not explored, the synergistic effects of SP and IGF-1C promoted cardiomyocyte survival, tissue neovascularization, and recruitment of stem/progenitor cells, providing a reparative response that obstructed the onset of myocardial fibrosis in infarcted hearts.

## 4. Prospect of Non-Mammalian Sourced Peptides as Anti-Fibrotics

In this review, we have provided a snapshot of recent investigations that have employed peptides as effective antifibrotic therapeutics. There continues to be strong interest in the research and development of peptides for this purpose, and nascent diversity in peptide candidate origins is becoming apparent. For example, studies on non-mammalian sourced peptides have yielded early evidence for untapped potential in controlling fibrotic processes: AMP Drs peptides (from *Drosophila* drosomycin) showed co-localization with F-actin and regulated collagen IV expression, thus having implications as an inhibitor of basement membrane aggregation in fibrosis [48]; and early studies on the PP20 peptide (PGSSVAVGVGKMKEAALAIV) derived from C-phycocyanin have indicated its negative regulation of the TGF-β1/Smad2/3 pathway [49]. A study by Xu et al. [50] explored the antioxidant and antifibrotic activity of the rapeseed protein-derived peptide RAP-8 (DHNNPQIR) in the context of liver fibrosis. The authors analyzed 588 overlapping and differentially expressed genes altered in RAP-8-treated mice with CCl_4_-induced liver fibrosis and identified cell cycle-associated genes (*Cenpp*, *Cyp2c55*, and *Serpinh1*) as relevant targets for treating liver fibrosis. RAP-8 was suggested to block liver fibrosis progression by inducing cell cycle arrest and preventing oxidative stress, and these mechanisms were verified in human hepatic stellate cells.

## 5. Discussion and Future Perspectives

Targeting fibrosis remains a scientific and medical challenge, but numerous studies have shown that blocking signaling pathways that drive fibrosis can inhibit the generation of myofibroblasts, offering directions towards antagonizing or reversing progressive organ fibrosis. Further in-depth understanding of the mechanisms of fibrosis and continuing breakthroughs in the research of interventions will be integral to preventing and treating fibrotic diseases. For example, the emergence and accessibility of single-cell transcriptomic analyses is likely to lead to the identification of distinct myofibroblast heterogeneity and fibroblast/myofibroblast subpopulations that may have more prominent roles in driving fibrotic processes in certain organs, which may in turn provide valuable insights as to how to better target antifibrotic peptides to key effector cells. In this review, we have provided a snapshot of recent investigations that have employed peptides as effective antifibrotic therapeutics. We focused on recent studies published within the last four years (after 1 January 2019) but this does not encompass all investigations involving peptides with demonstrable anti-fibrotic effects. There are several peptides undergoing investigation that have shown remarkable capabilities in preventing fibrosis. However, the precise mechanistic details have yet to be delineated; thus, a fair rationale for antifibrotic effects could not be accurately provided within the scope of this review. As with many small molecular drugs, some studies have shown other small peptides to have apparent anti-fibrotic effects, but without delineated mechanisms of action. Such observations, despite being superficial in details, are important starting points to warrant further exploration of the molecular mechanisms of action. Importantly, it is necessary to determine whether the inhibited fibrotic outcomes result from a depletion of myofibroblasts via stimulated apoptosis, the blockade of the TGF-β1 pathway/mechanotransduction components (at the level of ECM, receptors, intracellular signaling, or gene transcription and translation), or through the regulation of upstream initiators of auxiliary pro-fibrotic processes. Elucidation of which mechanism of action is the predominant effector would help in the design of targeted and timely therapeutics with desirable anti-fibrotic versus remodeling effects, and serve as a rationale for the development of new peptide-functionalized biomedicines.

An inherent advantage of peptides is their capacity to be readily modified and conjugated to potentiate and expand the function of either the peptide itself or the conjugated molecule. Such multiplicity was exemplified in an innovative study by Lee et al. [51], wherein the DPEA sequence of promelittin was substituted for the fibroblast activation protein (FAP)-sensitive GPA motif. Conjugation of the modified Cys-promelittin to maleimide-polyethylene glycol (mPEG) linked to the lipid anchor, 1,2-distearoyl-sn-glycero-3-phosphoethanolamine (DSPE), facilitated the decoration of liposomes with the FAP-responsive promelittin peptide. Subsequent delivery to multiple models of hepatic fibrosis demonstrated the localized release of the cytotoxic mature melittin in response to elevated FAP production by activated hepatic stellate cells (aHSC), thereby inducing aHSC apoptosis, reducing collagen accumulation, and enabling liver recovery. The study also helps to underline the issues related to peptide therapies administered without carrier materials, as highlighted in a review by Gu et al. [52], which provides a detailed account of the necessity for carrier materials to enhance kidney therapy efficacy. Much like drugs, the susceptibility of peptides to rapid degradation and unspecific uptake, whether by circulating cells or the cells of off-target organs, ultimately restricts efficacy and increases the risk of off-target effects and toxicity. An exemplary effort in prolonging the in vivo activity of a peptide was presented by Verdino et al. [53], wherein they conjugated single-chain RLX to the serum-albumin binding VHH domain to yield the relaxin analogue, LY3540378. The long-acting RLX analogue maintained selective receptor binding and activity comparable to native H2-relaxin hormone. LY3540378 treatments demonstrated attenuated hypertrophy in isoproterenol-induced cardiac hypertrophy mouse models. Thus, the concomitant development of pro-peptide (prodrug-like) systems, and peptide-loaded or peptide-based carrier materials is likely to form the focus of innovative studies going forward.

As with many of the peptides summarized in this review, peptides with desirable bioactivity often manifest from their larger protein counterparts. Inspiration can be taken from the application of truncated proteins with demonstrated anti-fibrotic activity, such as truncated TGFβRII [54] and truncated LAP [55]. While these larger protein studies are outside of the scope of this review, the design and development of smaller peptides based on the active domains of truncated proteins may yet yield cheaper alternatives with upscaled production prospects and improved stability in long-term storage that translate to being more readily employed as functionalization biomolecules for the generation of novel antifibrotic biomaterials. The co-development of technologies that can accurately model protein-peptide binding and predict interacting domains, such as deep-learning artificial intelligence (AI) [56] and diffusion modeling software [57], could result in the establishment of large databases containing protein structures with predicted binding sites for peptides to aid the design and testing of novel agonistic and antagonistic peptides. In our previous work, we utilized molecular docking to both identify the interaction site between EDA-FN and the VLA-4 integrin and to guide the design of polypeptides mimicking the integrin binding site with the intended function of competing with VLA-4 to bind and block EDA-FN-driven profibrotic cell activity [31]. In another example of in silico-aided peptide design, Mallart et al. [58] optimized the potency of the B-chain of H2-relaxin by minimalizing the structure to the active amino acids located at the 10–33 position before introducing specific mutations, suitable spacers, and fatty acid sidechains to produce long-acting single-chain peptide mimetics of H2-relaxin. The team used the Schrödinger tool suite (Release 2019-3: MacroModel, Schrödinger LLC, New York, NY, USA) to perform molecular dynamics simulations to rationalize gain-of-potency in their designed mimetics. The resultant peptide mimetics had sub-nanomolar activity, high bioavailability, and overcame the short half-life of H2-relaxin.

Most of the peptides reviewed here are in the preclinical stage of development and have yet to enter clinical trials. THR-184 and ANG-3777 (an HGF peptide mimetic previously known as BB3) have both completed phase 2 clinical trials (ClinicalTrials.gov ID: NCT01830920; NCT02771509) as perioperative treatments in cardiovascular-surgery-associated acute kidney injury (CVS-AKI), which is a type of ischemic injury that may progress to chronic kidney disease and renal fibrosis [59,60]. The E4 peptide is being developed as an E4-Fc fusion protein by iBio Inc. (San Diego, CA, USA) for scleroderma treatment, and they are reportedly applying for a new investigational drug application to the FDA to allow the commencement of phase 1 clinical trials. XFB-19 is a synthetic tetra-peptide (Acetyl-Lys-d-Ala-d-Val-Asp-NH_2_) that inhibits the activation of human CCAAT-enhancer binding protein beta (C/EBPβ). The nature of small peptides often limits their specificity for select cell types. However, in the case of XFB-19, this may be an advantage, as C/EBPβ has been suggested to have multiple antifibrotic activities, such as the regulation of macrophage polarity [61] and a role in the production of pro-inflammatory cytokines, pro-fibrotic ECM proteins, and α-SMA in cardiac fibroblasts/myofibroblasts [62]. XFB-19 is the only peptide with antifibrotic activity that has entered clinical trials in recent years (ClinicalTrials.gov ID: NCT05361733, accessed on 20 July 2023). Xfibra Inc. (Del Mar, CA, USA) is developing XFB-19 as a therapeutic for idiopathic pulmonary fibrosis and is currently enlisting subjects for their phase 1 human clinical trial, primarily aimed at assessing the safety, tolerability, and pharmacokinetics of single and multiple ascending doses of XFB-19 in healthy adult volunteers. The researchers working on the peptides reviewed here and new peptides being developed could look towards the E4 and XFB-19 research and developmental processes as exemplars for guidance towards entering phase 1 clinical trials. That is, the thorough investigation of their mechanisms of antifibrotic action in multiple cell types and in multiple preclinical models of organ fibrosis.

Undoubtedly, within the next few years of research on newly developed and existing peptides, as well as the results of clinical trials on peptide-based therapies for fibrosis, new answers will be revealed that help delineate mechanisms of therapeutic action while indicating issues that require attention. The results of these future investigations will add to the increasing pool of antifibrotic peptides and will provide a plethora of additional strategic routes towards the specific targeting, regulation, and prevention of the prolonged presence of myofibroblasts in progressive fibrosis and in multiple tissues.

## Figures and Tables

**Figure 1 biomolecules-13-01179-f001:**
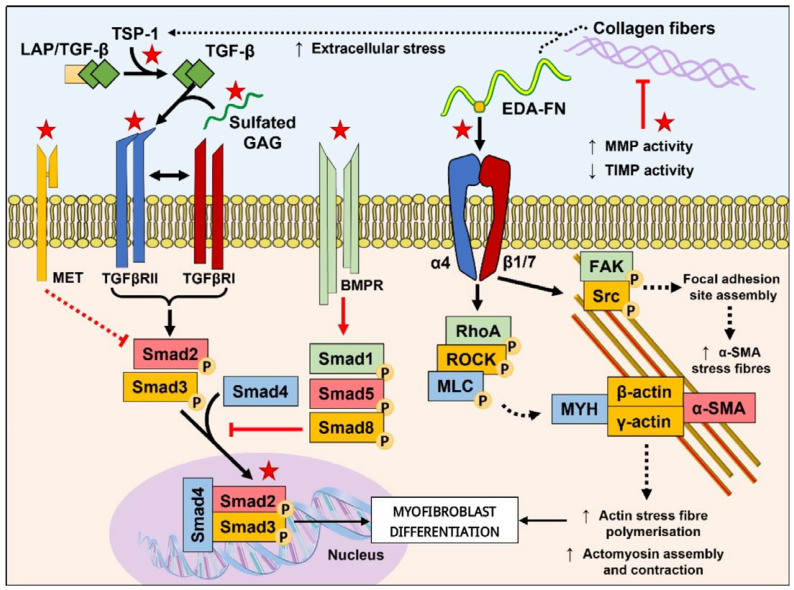
**Targeting the TGF-β/Smad and mechanotransduction pathways involved in driving myofibroblast differentiation.** The TGF-β/Smad pathway is the most well-studied pathway implicated in fibrosis and offers a range of mechanisms that have been targeted using peptides. Such interventions include the disrupted binding of TGF-β1 with its receptors through competing with sulfated GAGs to bind free TGF-β1 or direct inhibition of TGFβRII dimerization with TGFβRI; activation of antagonistic pathways such as the MET and BMPR pathways to regulate the activity of Smad2/3 phosphorylation and downstream activities; and the suppression of TGF-β1 driven Smad3 intracellular localization and transcriptional activity. The mechanotransduction activation of fibroblast to myofibroblast differentiation has become an increasingly popularized target for antifibrotic agents, including peptides. Interventions of mechanotransduction have been demonstrated through reducing matrix stiffness by targeting the urokinase and relaxin family peptide receptor-1 pathways that ultimately regulate collagen degradation through the upregulated expression of MMPs and decreased expression of TIMPs (which can in-turn regulate TGF-β release from the LAP complex); competing with TSP-1 recognition of LAP; and through the competitive binding of EDA-FN that would otherwise activate integrin-mediated mechanotransduction pathways and cytoskeleton remodeling. Red stars indicate targets for peptide intervention summarized in this review. Red lines, arrows, and dashed lines indicate inhibitory pathways; black lines, arrows, and dashed lines indicate pro-fibrotic pathways. Adapted and compiled with permission from figures in Tai et al. [4]. Abbreviations used in the diagram: TSP-1, thrombospondin-1; LAP/TGF-β, latency-associated peptide-TGF-β; GAG, glycosaminoglycan; EDA-FN, extra domain A-fibronectin; MMP, matrix metalloproteinase; TIMP, tissue inhibitor of metalloproteases; MET, mesenchymal-epithelial transition factor receptor; TGFβRII, TGFβ receptor type II; TGFβRI, TGFβ receptor type I; BMPR, bone morphogenetic protein receptor; α4, integrin α4; β1/7, integrin β1 and/or β7; FAK, focal adhesion kinase; Src, proto-oncogene tyrosine-protein kinase Src; RhoA, Ras homolog family member A; ROCK, Rho associated coiled-coil containing protein kinase; MLC, myosin light chain; MYH, myosin heavy chain; α-SMA, α-smooth muscle actin; Smad, mothers against decapentaplegic homolog.

**Table 1 biomolecules-13-01179-t001:** Emergent peptides and corresponding amino acid sequences in relevance to TGF-β pathway interference. dN = d-Asparagine; nV = Norvaline; dS = d-Serine.

Peptide	Origin/Inspiration	Amino Acid Sequence	Antifibrotic Mechanism	Ref.
CXCL9(74-103)	CXCL9	KKKQKNGKKHQKKKVLKVRKSQRSRQKKTT	Competitively binds heparan sulphate GAGs to inhibit its complexation with TGF-β1	[12]
KP1	Klotho	FQGTFPDGFLWAVGSAAYQTEGGWQQHGK	Competitively binds TGFβRII to inhibit TGFβRII/RI dimerization	[13]
C53	CNP	DLRVDTKSRAAWARLLQEHPNARKYKGANKKGLSKGCFGLKLDRIGSMSGLGC	Facilitates prolonged cGMP production, inhibits Smad3 nuclear translocation, and promotes Smad3 degradation	[7]
M10	MET	TRPASFWETS	Interacts with Smad2 to inhibit its nuclear translocation	[14]
H-RN	HGF	RNPRGEEGGPW	Inhibition of TGF-β2/Smad and Akt/mTOR signaling pathways	[15]
THR123	BMP7	CYFDDSSNVLCKKYRS	BMPR1A/Smad(1/5/(8)9) pathway agonist	[16]
THR184		CYYDNSSSVLCKRYRS		
ADP355	Adiponectin	H(dN)IP(nV)LY(dS)FA(dS)	Activates AMPK suppression of TGF-β1 induced transcription driven by Smad3-binding cis-elements	[17]

**Table 2 biomolecules-13-01179-t002:** Emergent peptides and corresponding amino acid sequences in relevance to ECM and mechanotransduction pathway interference.

Peptide	Origin /Inspiration	Amino Acid Sequence	Antifibrotic Mechanism	Ref.
B7-33	H2-Relaxin	VIKLSGRELVRAQIAISGMSTWSKRSL	RXFP1 agonist; promotes MMP2 expression and inhibits TIMP1, leading to reduced collagen fiber thickness	[30]
E4	Endostatin/COL18A1	SYCETWRTEAPSATGQASSLLGGRLLGQSAASCH HAYIVLCIENSFMT	Regulates the urokinase pathway and induces MMP-1/-3 degradation of ECM	[6]
AF38Pep	Integrin α4/β1 (VLA-4)	VMPYISTTPAKPCTSENCGNSWYGGFKSKNENKI YFIN	Competitively binds the EDA-FN C-C’ loop to prevent recognition by integrin α4/β1 or α4/β7	[31,32]
LSKL	LAP	LSKL	TSP-1 antagonist, suppresses the PI3K/Akt/mTOR pathways	[33]

## Data Availability

Data sharing not applicable.

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
