# Peer review of "Emergent Peptides of the Antifibrotic Arsenal: Taking Aim at Myofibroblast Promoting Pathways"

_biomolecules, 2023, doi:10.3390/biom13081179_

Round 1

Reviewer 1 Report

To the authors, a well-put-together mini-review that was clear and concise in its presentation. I have very little criticism or advice to the authors. 

A couple of small queries

 Page 4 end of second paragraph. It is written M10 was suggested to have the capability to reverse silica-induced endothelia-to-mesenchymal transition (EMT) in epithelial cells etc etc

Should this be epithelial-to-mesenchymal-transition (EMT)?

In the discussion it is stated:

“There are several peptides undergoing investigations that have shown remarkable capabilities in preventing fibrosis. However, the precise mechanistic details have yet to be delineated, thus, a fair rationale for antifibrotic effects could not be accurately provided within the scope of this review.”

Could this mini-review be expanded to include the author's professional opinion on the potential mechanisms of the antifibrotic effect(s)?

It would potentially provide valuable insight and topics for discussion within the relevant research community.

Author Response

Reviewer 1

To the authors, a well-put-together mini-review that was clear and concise in its presentation. I have very little criticism or advice to the authors. 

Response: We thank the reviewer for their positive comments. We have addressed the reviewer’s small queries, as detail below.

A couple of small queries

Page 4 end of second paragraph. It is written M10 was suggested to have the capability to reverse silica-induced endothelia-to-mesenchymal transition (EMT) in epithelial cells etc etc. Should this be epithelial-to-mesenchymal-transition (EMT)?

Response: The reviewer is correct. We apologise for this careless oversight and thank the reviewer for spotting the mistake. We have corrected it to epithelial-to-mesenchymal transition.

In the discussion it is stated:

There are several peptides undergoing investigations that have shown remarkable capabilities in preventing fibrosis. However, the precise mechanistic details have yet to be delineated, thus, a fair rationale for antifibrotic effects could not be accurately provided within the scope of this review.

Could this mini-review be expanded to include the author's professional opinion on the potential mechanisms of the antifibrotic effect(s)? It would potentially provide valuable insight and topics for discussion within the relevant research community.

Response: As with many small molecular drugs, studies have shown small peptides that appear to exhibit anti-fibrotic effects but without delineated mechanisms of action. Such observations, despite being superficial in details, are important starting points to warrant the further exploration of the molecular mechanisms of action. Importantly, it is necessary to determine whether the inhibited fibrotic outcomes are resultant from a depletion of myofibroblasts via stimulated apoptosis, the blockade of the TGF-β1/mechanotransduction components (at the level of ECM, receptors, intracellular signalling, or gene transcription and translation), or through the regulation of upstream initiators of pro-fibrotic processes. Elucidation of which mechanism of action is the predominant effector would help in the design of targeted and timely therapeutics, and serve as a rationale basis for the development of new peptide functionalised biomedicines.

We thank the reviewer for their comment, although the extent of the conversation around the topic is brief in nature, we have added this opinion after the above indicated statement in the discussion in the hope that it may guide readers to check the essential mechanisms as a starting point.

Reviewer 2 Report

This is a timely and well-written review. My only request is that the authors be more thorough in define the abbreviations used.

Introduction, 1st paragraph. What is EDA? 

Figure 1 contains a number of abbreviations, these should be defined in the Figure legend. 

There are formating problems with the Figures that need to be fixed. Please keep Table 1 together on one page.

Author Response

Reviewer 2

This is a timely and well-written review. My only request is that the authors be more thorough in define the abbreviations used. Introduction, 1st paragraph. What is EDA? Figure 1 contains a number of abbreviations, these should be defined in the Figure legend. 

Response: We thank the reviewer for their positive comments and indicating areas where our manuscript can be improved. We have followed the reviewer’s advice and provided the long form names of all abbreviations in their first instances, and in the figure legend.

There are formatting problems with the Figures that need to be fixed. Please keep Table 1 together on one page.

Response: Our apologies for the formatting errors, we have rearranged the Figure and Table to ensure it fits on one page.

Reviewer 3 Report

Zhen Liu et al. in this systematic review describe emergent antifibrotic peptides and their utilization on myofibroblasts, peptide inhibitors of upstream pathogenic processes that drive the formation of profibrotic cell phenotypes. The manuscript has its merits, and is interesting; however, they have some minor and major corrections.

It would be advisable that Figure 1 be better explained in the text or in the foot figure, as it has a lot of content and there is not an adequate explanation.

It is recommended that Table 1 appears at the end of 2.1, since it is difficult to understand at the beginning and becomes a little more comprehensible at the end.

The authors mention the mechanism. It would be essential to mention who forms the peptides, i.e. who hydrolyzes the proteins (CXCL, FGF23,...).

It is suggested that the authors add a separate section on perspectives and expectations, found in section 5.

Mention in a section if there are peptides that are in clinical phases.

Author Response

Reviewer 3

Zhen Liu et al. in this systematic review describe emergent antifibrotic peptides and their utilization on myofibroblasts, peptide inhibitors of upstream pathogenic processes that drive the formation of profibrotic cell phenotypes. The manuscript has its merits, and is interesting; however, they have some minor and major corrections.

It would be advisable that Figure 1 be better explained in the text or in the foot figure, as it has a lot of content and there is not an adequate explanation.

Response: Our thanks for this helpful suggestion, we have supplemented the Figure Legend with additional text to clarify the image, additionally, we added full terms for the abbreviations used.

It is recommended that Table 1 appears at the end of 2.1, since it is difficult to understand at the beginning and becomes a little more comprehensible at the end.

Response: We have followed the reviewer’s advice and rearranged the tables to appear at the end of their relevant sections.

The authors mention the mechanism. It would be essential to mention who forms the peptides, i.e. who hydrolyzes the proteins (CXCL, FGF23,...).

Response: The peptides used in those studies are synthetic. Whilst truncated forms of the protein of origin exist in human cells, the researchers utilized optimized synthetic peptides to demonstrate the high affinity binding and antifibrotic effects. We edited the text to clarify the synthetic production of the CXCL9(74-103) and KP1 proteins inspired by the observations made about CXCL9 COOH-terminal and the highly conserved 30 amino acid region of Klotho-KL1 domain, respectively.

It is suggested that the authors add a separate section on perspectives and expectations, found in section 5.

Response: We have added additional perspectives on the future requirements and recommendations for the design of studies to accelerate to the discovery of antifibrotic peptides as part of section 5.

Mention in a section if there are peptides that are in clinical phases.

Response: The inclusion of peptides was dependent on their featuring in recent studies, as such, none of the reviewed peptides have entered clinical trials (as per ClinicalTrials.gov). That being said, it was noted that E4 is expected to enter phase 1 clinical trials pending an IND application to the FDA. In section 5 on future perspectives, we include a section that discusses an exemplar peptide candidate (XFB-19) and its progression toward safety and tolerance phase 1 trials in healthy humans. We also recommend that researchers look toward E4 and XFB-19 research and development as guides on how to prepare novel peptide products for potential entrance into clinical trials.

Reviewer 4 Report

There is nothing inherently wrong with the review in that it is successful at describing its topic.

However, there is not a critical evaluation. Essentially all of these peptides (perhaps not? I may be mistaken?) are at the discovery/basic research stage and are not in the clinic.  Moreover, the targets to which the peptides are raised are not necessarily validated. Can the authors please discuss what remains to be done to each peptide to get it into the clinic and the stage of clinical development for each peptide? A table would be useful.

Author Response

Reviewer 4

There is nothing inherently wrong with the review in that it is successful at describing its topic.

However, there is not a critical evaluation. Essentially all of these peptides (perhaps not? I may be mistaken?) are at the discovery/basic research stage and are not in the clinic.  Moreover, the targets to which the peptides are raised are not necessarily validated. Can the authors please discuss what remains to be done to each peptide to get it into the clinic and the stage of clinical development for each peptide? A table would be useful.

Response: We thank the reviewer for their positive comments. The inclusion of peptides was dependent on their featuring in recent studies, as such, none of the reviewed peptides have entered clinical trials (as per ClinicalTrials.gov). That being said, it was noted that E4 is expected to enter phase 1 clinical trials pending an IND application to the FDA. In section 5 on future perspectives, we include a section that discusses an exemplar peptide candidate (XFB-19) and its progression toward safety and tolerance phase 1 trials in healthy humans. We also recommend that researchers look toward E4 and XFB-19 research and development as guides on how to prepare novel peptide products for potential entrance into clinical trials.

Round 2

Reviewer 3 Report

All suggestions and comments were adequately addressed by the authors.

Author Response

Please see uploaded revised manuscript